# Study on the Source and Microbial Mechanisms Influencing Heavy Metals and Nutrients in a Subtropical Deep-Water Reservoir

**DOI:** 10.3390/microorganisms13122750

**Published:** 2025-12-03

**Authors:** Gaoyang Cui, Jiaoyan Cui, Mengke Zhang, Boning Zhang, Yingying Huang, Yiheng Wang, Wanfu Feng, Jiliang Zhou, Yong Liu, Tao Li

**Affiliations:** 1Faculty of Geographical Science and Engineering, College of Geographical Science, Henan University, Zhengzhou 450046, China; aycgy@henu.edu.cn (G.C.);; 2Henan Dabieshan National Field Observation and Research Station of Forest Ecosystem, Zhengzhou 450046, China; 3Xinyang Academy of Ecological Research, Xinyang 464000, China; 4The Forest Science Research Institute of Xinyang, Xinyang 464031, China; 5Henan Jigongshan Forest Ecosystem National Observation and Research Station, Xinyang 464031, China

**Keywords:** nutrients, heavy metals, microorganisms, interaction, stratified reservoir

## Abstract

Reservoirs are hotspots for the coupling of nutrients and heavy metals, and they substantially modify the compositions and spatiotemporal distributions of microorganisms in fluvial systems. However, relatively few studies have been performed that investigate the microbial mechanisms driving interactions among heavy metals and nutrients in reservoirs. The Goupitan Reservoir, a seasonal stratified reservoir located within the Wujiang River catchment, was chosen as the research subject. The temporal and spatial variations in heavy metals and nutrients, and the metagenomic composition of the reservoir water were analyzed in January, April, July, and October 2019. The results revealed that As, Ni, Co, and Mn were derived primarily from mine wastewater, whereas Zn, Pb, Cd, and Cr were related to domestic and agricultural wastewater discharge. The study area was dominated by *Proteobacteria*, *Actinobacteria*, *Cyanobacteria*, and *Bacteroidetes*, with the proportion of dominant phyla reaching 90%. Decreases in the dissolved oxygen (DO) concentration and pH in the bottom water during July and October were conducive to increases in the abundance of the anaerobic bacterial groups *Planctomycetes* and *Acidobacteria*. The functional genes *norBC* and *nosZ* associated with denitrification (DNF), the key gene *nrfAH* involved in the dissimilatory nitrate reduction to ammonium (DNRA) process, the functional genes *aprAB* and *dsrAB* responsible for sulfate reduction/sulfide oxidation, as well as the thiosulfate oxidation complex enzyme system *SOX*, all exhibit high abundance in hypoxic water bodies and peak in the redoxcline, highlighting the significance of related nitrogen (N) and sulfur (S) metabolic processes. In addition, the concentrations of heavy metals significantly affected the spatial differentiation of the planktonic bacterial community structure, with Mn, Co, Fe, Ni, As, and Cu making relatively high individual contributions (*p* < 0.01). This study is important for elucidating the sources and microbiological mechanisms influencing heavy metals and nutrients in seasonally stratified subtropical reservoirs.

## 1. Introduction

Eutrophication, the enrichment of heavy metals, and their combined effects are typical water pollution issues that have attracted widespread attention from scholars both in China and other countries [1,2,3]. The excessive accumulation of nutrients such as nitrogen (N) and phosphorus (P) in water bodies due to inputs from weathering products, industrial and agricultural production activities, and domestic sewage discharge can lead to an imbalance in a system’s nutritional structure, disrupting the existing species balance and ecology [4,5]. Heavy metals from natural sources, such as rock and mineral weathering, as well as anthropogenic sources such as mining, industry, agriculture, and domestic waste, can not only disrupt the balance of aquatic ecosystems but also harm human health through bioaccumulation effects and direct ingestion [6,7]. To date, relatively comprehensive research on source analysis, evaluation systems, and control measures has been independently conducted for both eutrophication and heavy metal pollution in water bodies [3,8]. However, as water environmental issues become increasingly complex, there is an urgent need to explore the microbiological mechanisms influencing combined pollution from eutrophication and an excess of heavy metals [9].

Aquatic organisms, particulate matter, and organic macromolecules are common carriers of heavy metals and nutrients [10,11]. Eutrophication directly accelerates the proliferation of aquatic algae, and algae’s photosynthesis leads to relative oxygen enrichment in surface waters. The bottom water becomes oxygen-deficient because of the oxygen–consuming decomposition of dead algal bodies that settle to the bottom. Furthermore, changes in redox conditions at the bottom affect the pH of the water body [5,11]. Moreover, changes in water temperature, pH, DO, etc., caused by eutrophication further affect the speciation of heavy metals [8,9]. Studies have shown that there is some antagonistic relationship between the level of water eutrophication and the degree of heavy metal pollution [9]. Seasonal stratification in reservoirs leads to changes in the physicochemical properties of water, which in turn affect the forms and distribution processes of nutrients and heavy metals [12,13]. Currently, domestic and international research on the coupling mechanisms of heavy metals and nutrients in water bodies primarily centers on employing stable isotopes or microbial genomics techniques to investigate individual coupling processes occurring in anoxic marine sediments, river sediments, deep-sea hydrothermal vents, and wastewater treatment systems [14,15,16]. However, our understanding of the interaction mechanisms among electron donors, microbial populations, and functional genes during the coupled processes of heavy metal and nutrient cycling in seasonally stratified deep-water reservoirs remains insufficient.

In this study, the Goupitan Reservoir, which is a seasonally stratified deep-water reservoir and is located in the main stream of the Wujiang River, was selected as the research object. The research objectives of this paper are as follows: (1) Clarify the spatiotemporal distribution patterns of nutrients and heavy metal concentrations in Goupitan Reservoir, emphasizing the disparities in the redox processes of elements between aerobic and anaerobic conditions, and investigate the geochemical cycling mechanisms of elements in the reservoir’s water under seasonal hypoxic conditions; (2) reveal the spatiotemporal distribution patterns of microbial composition and the relative abundance of functional genes in Goupitan Reservoir, with particular emphasis on seasonal variations as well as differences between anoxic and aerobic water bodies during stratification periods. Investigate the relationship between alterations in microbial composition and function and the spatiotemporal distribution of nutrients and heavy metals, and identify the microbial-driven mechanisms underlying the coupled nutrient-heavy metal cycling processes in seasonally stratified reservoir waters. By investigating the sources and microbial processes influencing heavy metals and nutrients in this area, this study provides a theoretical basis for exploring the mechanism of coupling among heavy metals and nutrients. Additionally, this study offers guidance for addressing combined pollution issues in karst deep reservoirs.

## 2. Materials and Methods

### 2.1. Study Area

As a southern tributary of the Changjiang River, the Wujiang River originates in the Wumeng Ranges, extends more than 1037 km, drains 8.8 × 10^4^ km^2^, and has an average runoff of 53.4 × 10^9^ m^3^ [17]. The elevation ranges from 500 to 1500 m above sea level. The annual precipitation varies between 850 and 1600 mm, with rainfall from May to October constituting 80% of the total annual precipitation. The power generating capacity in the basin is 10.4259 million kW, of which 5.804 million kW are in the main stream of the Wujiang River.

The Goupitan Reservoir, located in Yuqing County, Guizhou Province, has functions such as power generation, shipping, and flood control. It represents the fifth cascade development stage of the mainstream of the Wujiang River. With a total storage capacity of 6.454 billion cubic meters and a regulating storage capacity of 2.902 billion cubic meters, the reservoir has a normal water level of 630 m and controls a drainage area of 43,250 square kilometers. It is a typical deep–water lake reservoir with the greatest hydropower development intensity in the Wujiang River Basin. The sewage outfall upstream of the Goupitan Reservoir has contributed to its high phosphorus background value. However, in recent years, the flocculation–based phosphorus removal facilities at the end of the Yangshui River tributary and at No. 34 Spring have enabled the total phosphorus concentration to be controlled within the range of 0.19–0.30 mg·L^−1^ except during the rainy season. This has led to severe pollution issues in most of the main tributaries of the Goupitan Reservoir.

### 2.2. Analyzing and Sampling

Sampling was conducted in January, April, July, and October of 2019, representing the dry, flood, and normal seasons, respectively. The sampling area included tributaries (Yangshui River, Xiangjiang River, Datang River, Wengan River, and Qingshui River), stratified water in the reservoir area (27°38′19″ N; 107°64′47″ E), and discharged water (Figure 1). Only surface water was taken from the tributaries and discharged into water bodies (27°37′10″ N; 107°65′11″ E). For the profile water in the reservoir area, sampling was conducted using a Niskin water sampler from the General Oceanics Company in Miami, FL, USA, starting from the bottom, with intervals of 10 or 20 m for the middle and lower parts (>30 m) and 5 m for the upper part (<30 m).

Water temperature, DO, pH, total dissolved solids (TDS), oxidation-reduction potential (ORP), and chlorophyll *a* (Chl *a*) were measured in situ using an automated multiparameter profiler (YSI 6600, Xylem, Washington, DC, USA). After water sampling, the water was filtered through a Sartorius Teflon filter using a 0.45 μm mixed fiber filter membrane (Millipore, Burlington, MA, USA, diameter 142 mm) within 6 h [18]. The heavy metal samples were subjected to secondary nitric acid acidification (pH < 2). The water sample used for measuring nutrients is filtered and fixed with chloroform, then refrigerated at 4 °C and the nitrate concentration was determined by cadmium copper reduction method, the nitrite concentration was determined by diazo azo method, the ammonia nitrogen concentration was determined by indigo phenol blue method, and the silicate concentration was determined by silicon molybdenum blue method, total dissolved nitrogen (TDN) concentration determined by potassium persulfate oxidation colorimetric method [19]. The concentrations of nutrients (TDN, NH_4_^+^, NO_3_^−^, NO_2_^−^, Total Dissolved Phosphorus—TDP, PO_4_^3−^, and Dissolved Silicon—DSi) in the water were measured using a Skalar San++ continuous flow analyzer from Breda, The Netherlands. The concentrations of heavy metals were determined by inductively coupled plasma mass spectrometry (ICP-MS8900, Agilent, Santa Clara, CA, USA) using the ^103^Rh single internal standard online addition method [6]. The detection limits of this instrument for elements ^52^Cr, ^55^Mn, ^56^Fe, ^59^Co, ^60^Ni, ^63^Cu, ^66^Zn, ^75^As, ^111^Cd, and ^208^Pb are 1.39 × 10^−2^ μg·L^−1^, 2.78 × 10^−2^ μg·L^−1^, 3.99 × 10^−2^ μg·L^−1^, 2.06 × 10^−3^ μg·L^−1^, 3.96 × 10^−2^ μg·L^−1^, 6.17 × 10^−3^ μg·L^−1^, 6.52 × 10^−2^ μg·L^−1^, 7.51 × 10^−3^ μg·L^−1^, 3.41 × 10^−4^ μg·L^−1^ and 9.65 × 10^−4^ μg·L^−1^, respectively, the recovery rate of the internal standard is 80.76%–116.00%. The concentration of dissolved organic nitrogen (DON) was calculated by subtracting inorganic nitrogen (including NO_3_^−^–N, NO_2_^−^–N, and NH_4_^+^–N) from total dissolved nitrogen (TDN). The relative deviation of the sample results was less than 5%; all standard curves had a correlation coefficient of more than 99.9% [20].

The microbial samples were filtered using 0.22 µm acetate cellulose filter membranes that had been sterilized under high temperature and high pressure (121 °C, 30 min). Generally, 500 mL of the water sample was filtered through each membrane until a visible coating was observed on the membrane. Six membranes were used for filtration and stored for future use. The membrane samples were immediately placed in a −20 °C refrigerator for cryogenic storage.

DNA extraction was carried out using the E.Z.N.A.® Soil Kit produced by Omega Bio-tek in Norcross, GA, USA, following the corresponding steps outlined in its instructions to extract total DNA from the filter membrane. After DNA extraction, the concentration of the extracted DNA was measured using a NanoDrop2000 (Thermo Fisher Scientific, Wilmington, DE, USA), and then the quality of the DNA obtained in the experiment was verified using a 1% agarose gel electrophoresis method, resulting in corresponding DNA electrophoresis bands. PCR amplification was performed on qualified DNA, selecting primers 338F/806R targeting the V3–V4 variable region of the DNA for the amplification process. The amplification was conducted using an ABI GeneAmp® 9700 PCR instrument (Foster City, CA, USA).

Recover the amplified PCR product using a 2% agarose gel, then purify the product with the AxyPrep DNA Gel Extraction Kit from Axygen Biosciences (Union City, CA, USA), elute the PCR product with Tris-HCl reagent, and perform electrophoresis using a 2% agarose gel to assess the quality of the amplified product. Finally, conduct quantitative detection of the purified PCR products using the QuantiFluor™-ST reagent from Promega Corporation (Madison, WI, USA). Subsequently, construct a PE 2*300 library from the purified amplification products following the standard protocol recommended by the Illumina MiSeq platform (San Diego, CA, USA). The raw data obtained from sequencing were uploaded to the NCBI database, and the SRA sequence number for this study is SRP321486. Using UPARSE software version 7.1, initiate the OTU clustering process on the quality-controlled sequences according to the conventional 97% similarity threshold, and eliminate chimeras and singleton sequences during this process. Subsequently, employ the RDP classifier (available at: http://rdp.cme.msu.edu/, accessed on 26 November 2025) to perform hierarchical species classification annotations on each obtained sequence. Finally, compare these sequences against the Silva SSU123 database using a 70% similarity threshold to obtain species annotation results. For the annotation results, we employed one-way ANOVA (suitable for three or more sample groups) to assess whether there were significant differences in species composition across different groups. The FDR multiple testing correction method was applied to adjust the *p*-values, and post hoc tests were conducted to analyze the species showing differences, thereby identifying the sample groups with significant variations among the multiple groups. The sequencing data involved in this study were processed on the Meiji Cloud platform (specific URL: https://cloud.majorbio.com/, accessed on 26 November 2025). 

Metagenomic sequencing was conducted using the Illumina HiSeq Xten sequencing platform from the United States (Shanghai Meiji Biomedical Technology Co., Ltd., Shanghai, China). The specific process unfolds as follows: Initially, corresponding template information is immobilized on the chip through amplification, with one end exhibiting complementarity to the bases in the primer. The other end of the template molecule is randomly fixed by forming a complementary bond with another surrounding primer, thereby creating a bridge. Subsequently, the template molecules undergo PCR amplification to generate DNA clusters. Following this, the DNA amplicons are linearized to form single-stranded DNA. Next, by introducing modified DNA polymerase and dNTPs labeled with four distinct fluorescent markers, one base is synthesized per cycle. The types of nucleotides clustered on the template sequence after the initial reaction round are then identified (using laser scanning to examine the surface of the reaction plate). The 3′ end’s reactivity is restored by cleaving the “fluorescent” and “terminating” groups, initiating the polymerization of the next nucleotide. Finally, the fluorescence signals acquired from each cycle are statistically processed to derive the corresponding sequence information of the template DNA fragment. The raw data of the metagenomic sequences of the water samples in this study have been uploaded to the GenBank sequence database with the login number PRJNA799263. We utilized Diamond software version 0.8.35 to compare the amino acid sequences of nonredundant gene sets with those in the corresponding NR database. During alignment, the parameter setting for BLASTP was set to an e-value of 1e−5. We calculated the relative abundance of different functions by summing the gene abundances corresponding to the KO (KEGG orthology) values, metabolic pathways (Pathway), enzyme composition (EC), and metabolic modules (Module). The experimental data processing was completed using Microsoft Excel 2013; using Origin 2017 and CorelDRAW for related drawing work; performing correlation analysis and significance test on various variables of the data using SPSS 25.0 (IBM, Armonk, NY, USA) (*p* ˂ 0.05).

## 3. Results

### 3.1. Physicochemical Properties and Chlorophyll A

The sampled water body was alkaline, with a pH range of 7.36–9.45. The dissolved oxygen (DO) content of the water body is highest in April (6.78–20.02 mg·L^−1^, with an average of 8.33 mg·L^−1^) and lowest in October (1.46–9.23 mg·L^−1^, with an average of 4.99 mg·L^−1^). The chlorophyll *a* (Chl *a*) content of the water body was relatively high in April and July, reaching a maximum of 46.58 µg·L^−1^. In July, the water body in the study area exhibited a distinct thermocline (Figure 2). The overall trends in temperature (T), dissolved oxygen (DO), and pH in the water profile were as follows: they reached their maximum at the surface, then steeply decreased with depth to a specific layer and stabilized, conforming to the single-cycle pattern of seasonal stratification in deep–water reservoirs. In April and July, the surface water reached a supersaturated state for DO (>8 mg·L^−1^), accompanied by extremely high Chl *a* content, indicating strong photosynthesis by phytoplankton in the surface water during these seasons.

### 3.2. Temporal and Spatial Variations in Water Nutrients

The mean TDN content in the study area was 4.47 mg·L^−1^, with NO_3_^−^–N being the primary form of TDN in the water body (Appendix A). The concentrations of NO_3_^−^–N and NH_4_^+^–N in the surface water of the reservoir area in April and July were significantly higher than those in January, whereas PO_4_^3−^ was not evident. The Yangshui River, a phosphorus mine discharge point, had a generally high TDP concentration (0.16–0.50 mg·L^−1^), and the DSi concentration at this location was the highest. In the profile of the reservoir area, strong biological activity in the surface layer (0–10 m) led to a lower nutrient salt content, and as the water depth increased, the nutrient salt content gradually increased.

### 3.3. Temporal and Spatial Variations in Heavy Metals

The concentrations of heavy metals in the water of the Goupitan Reservoir, except for arsenic (As), were significantly lower than the recommended values specified in the “Drinking Water Hygiene Standards” (GB5749-2006) and the World Health Organization’s guidelines (2011) [21,22]. The arsenic concentration in the Yangshui River ranged from 7.32 to 9.41 μg·L^−1^, which is close to the recommended health risk threshold (10 μg·L^−1^). The average concentrations of soluble heavy metal ions in the water were in the following order: Fe > Zn > Mn, As, Ni > Cu, Cr > Pb > Co > Cd (Appendix A). In April, July, and October, the overall trend in the heavy metal ion concentration was as follows: middle layer > bottom layer > surface layer (Figure 3).

### 3.4. Temporal and Spatial Variation in Microbial Communities

The 16S rRNA gene in 131 samples, including sediment, from the reservoir area was sequenced using Illumina MiSeq, with read lengths ≥ 200 bp. After quality filtering with QIIME, 6,209,877 high–quality reads were obtained. The total sequences were assigned to 64 phyla, 151 classes, 433 orders, 764 families, 1706 genera, 3982 species, and 14,435 OTUs.

At the phylum level, the study area harbored a total of 64 known bacterial phyla across the four seasons (Figure 4). *Proteobacteria*, *Actinobacteria*, *Cyanobacteria*, and *Bacteroidetes* were the predominant phyla, accounting for 90% or more of the total. There were some differences in community composition at the phylum level across different seasons and profiles. This was reflected mainly in significant changes in the relative abundance of bacteria at the phylum level in July and October. In particular, the proportions of the dominant phylum *Bacteroidetes* and nondominant phyla, such as *Planctomycetes* and *Chloroflexi*, increased significantly in July and October. Additionally, the abundance of *Firmicutes* decreased significantly in July, whereas that of *Acidobacteria* increased significantly.

## 4. Discussion

### 4.1. Source Analysis of Heavy Metals

The results of cluster analysis revealed that As, Ni, Co, and Mn concentrations were strongly correlated (Figure 5). These elements are influenced not only by urban emissions, pesticide and fertilizer residues from surrounding areas, and mine wastewater, but also by the decomposition of sediments and dead organisms [5,6,7]. Elements such as Zn, Pb, Cd, and Cr are primarily associated with industrial, domestic, and agricultural wastewater and traffic emissions [7]. Moreover, the high levels of Fe and Cu in water bodies are attributed mainly to natural sources, primarily natural background inputs [6]. The concentrations of the aforementioned elements were also high in the downstream Yangshui River due to the presence of phosphate mines, confirming that the primary source of these elements was mine wastewater. Typically, when ligands chelated with Co and Mn–Fe oxides react with sulfides, Co is released [23]. The concentration of Ni is related primarily to the degradation of organic matter, but Ni may also be bound to iron oxides [23]. As exists mainly in the forms of sulfides and iron sulfides, and it is also influenced by the discharge of mine wastewater [24].

There was also a highly significant positive correlation between Zn and Pb (Zn vs. Pb, *r* = 0.66, *p* < 0.01) and between Zn and Cu (Zn vs. Cu, *r* = 0.64, *p* < 0.01; Figure 6), indicating that they may originate primarily from domestic and agricultural sewage discharge and are affected by the same factors [6]. This has also been confirmed in the Weng’an River and Qingshui River (Appendix A). Cu often complexes with organic matter and exhibits strong biological affinity. Under anaerobic conditions, the decomposition of organic matter and dead organisms can lead to an increase in Cu concentration [7], Zn is less affected by redox conditions, and its apparent peak values observed in April and July are most likely due to the combined effects of industrial and agricultural wastewater discharge, as well as the oxidative release of pollutants stored in sediments from past cage aquaculture under anaerobic conditions [23]. Like Zn, Pb is affected by sediment, but it is also affected by anthropogenic sources such as traffic [6]. In this study, the distribution of Cr is similar to that of Fe (*r* = 0.59, *p* < 0.01), and previous studies have reported that chromium oxide may be bound to iron oxides [24]. Cd concentrations, on the other hand, are simultaneously controlled by multiple processes, such as the adsorption and desorption of organic matter and biological particles in water, as well as sulfide formation, and therefore do not exhibit a clear pattern of variation with depth [24].

### 4.2. Microbial Mechanisms Influencing Heavy Metals

#### 4.2.1. Environmental Factors Influencing Heavy Metals and Microorganisms

This study explored the specific environmental factors that affect the composition of planktonic bacterial communities through RDA analysis graphs, including hydrochemical indicators ORP, DO, Chl a, nutrients NH_4_^+^ N, DOC, PO_4_^3−^, SO_4_^2−^, as well as heavy metals Mn, Co, Fe, Ni, As, Cu, etc., and their effects on the composition of planktonic bacteria in Goupitan Reservoir. ORP was significantly positively correlated with TDP, DSi, and DON concentrations, whereas DO was significantly negatively correlated with TDP, DSi, and DON concentrations (*p* < 0.01; Figure 7). This may be because high ORP and low DO, i.e., anoxic reduction conditions, promote the decomposition of particulate phosphorus, dead organisms, and sediment organic matter in the water body [9]. ORP was significantly negatively correlated with NO_3_^−^–N and NH_4_^+^–N concentrations, as low ORP promotes the denitrification reaction. Chl *a* was significantly negatively correlated with TDN, TDP, DSi, and PO_4_^3−^ concentrations, which also confirms that strong biological activity in the surface water body leads to enhanced assimilation and absorption of nutrients.

The decrease in DO concentration in the bottom water in July and October was conducive to the growth and reproduction of the anaerobic bacteria group *Planctomycetes*. The decrease in the pH in the bottom water in July was more conducive to the survival of *acidophilic microorganisms*, and the increase in *Acidobacteria* is a response of the microbial community to the decrease in water pH. Therefore, seasonal changes in the composition of microbial communities can also reflect, to some extent, the changes in water chemistry. Mn mainly showed a significant negative correlation with pH (Mn vs. pH, *r* = −0.357, *p* < 0.01), as low pH values are conducive to the release of particulate Mn [19], and Fe mainly showed a significant negative correlation with Chl and T (Fe vs. T, *r* = −0.357, *p* < 0.01; Fe vs. Chl *a*, *r* = −0.408, *p* < 0.01), indicating that strong biological assimilation and absorption in the surface layer lead to a decrease in Fe content [20,23].

#### 4.2.2. Mechanism of Coupling Between Heavy Metals, Nutrients, and Microorganisms in Reservoirs

The concentrations of Fe, Co, Ni, and As were correlated with the concentration of PO_4_^3−^ (the correlation coefficients were all greater than 0.249, *p* < 0.05, and *n* = 84, indicating a moderate correlation), demonstrating that the sources and geochemical behaviors of these heavy metals are consistent with those of PO_4_^3−^, and are primarily influenced by surface runoff and the release of particulate pollutants in water, which are consistent with P sources [16]. There was also a significant positive correlation between the concentrations of Fe, Co, Ni, As, Mn, and DSi (the correlation coefficients range from 0.405 to 0.671, *p* < 0.01, *n* = 84), indicating that runoff and biological activities (biological assimilation and absorption, as well as the decomposition of diatoms and other organisms) closely related to DSi concentration also affected the geochemical behavior of heavy metals in the water. Therefore, the concentration of heavy metals in water bodies was influenced primarily by nutrients such as PO_4_^3−^ and DSi. Runoff input played a significant role in the coupling between heavy metals and nutrients in the water bodies. The adsorption/desorption processes of particulate pollutants and biological interactions also affected this complex coupling process.

Existing research has indicated that the larger the DSi/TDN ratio is, the more pollutants are derived mainly from internal sources [9]. In this study, the DSi/TDN values ranged from 0.10 to 1.96, with an average of 0.61, indicating that internal sources also played a crucial role. Based on the geochemical properties of metals such as Fe, Zn, and As, it is speculated that the higher the As/Fe ratio is, the greater the role of sulfate reduction in the system, and that the higher the Zn/As ratio is, the more important the release process of particulate pollutants becomes [23]. Combined with the scatter plot of characteristic elemental ratios, it is believed that the coupling relationship between heavy metals and nutrients in this region is influenced primarily by various factors, such as runoff, sulfate reduction, and the release of particulate pollutants (Figure 6b).

In July and October, the bottom samples were positively correlated with NH_4_^+^–N, DOC, and PO_4_^3−^, indicating that the hypoxic environment and nutrients in the bottom layer of the Goupitan Reservoir are crucial for the proliferation of anaerobic bacterial communities (Figure 7a). Ávila et al. also reported that in seasonally stratified lakes, differences in bacterial richness and composition may be closely related to the physicochemical characteristics of the water column, especially PO_4_^3−^, pH, and dissolved oxygen (DO). Microbial composition is extremely sensitive to environmental changes. As seasonal stratification occurs in water bodies, notable disparities emerge in their physicochemical indicators, and the overall microbial makeup of water varies across different seasons and depths. Based on hierarchical clustering analysis of the samples, all samples from the study area can be roughly categorized into six groups (Appendix A). The degree of similarity or genetic relationship between different samples can be assessed by examining the distances between them. All samples collected in January fall into a single category. In April, samples from depths greater than 4 meters are grouped into one category, with the dominant microbial species potentially linked to photosynthesis. Samples from depths less than 4 meters form a distinct subgroup. In July, samples from depths of 16 meters and below, along with those from 18 meters and below in October, are categorized together, likely influenced by hypoxic conditions, with the dominant bacterial populations affected by the distribution of dissolved oxygen. Samples from depths greater than 16 meters in July are classified separately as one category, while samples from 0 to 18 meters in October are grouped into another category (Appendix A). Stratification and hypoxia during the rainy season promote the emergence of hypoxic photosynthetic and *methanotrophic bacteria* (Appendix A), which are important for carbon and nutrient cycling. As shown in Appendix A, the top 6 genera with the highest content are *Sulfuritalea*, *unclassified_f_Thiobacillaceae*, *unclassified_f_Rhodocyclaceae*, *unclassified_o_Rhodocyclales*, *unclassified_o_Nitrosomonadales*, and *Geobacter*. *Sulfuritalia*, *unclassifiedd_f_Thiobacillaceae*, *unclassifiedd_o’Nitrosomonaales* belong to *Nitrosomonaales* (order *Methylomonas*), which are a class of chemoautotrophic microorganisms capable of oxidizing sulfides, thiosulfates, and elemental sulfur and reducing nitrates. In July and October, the bottom samples were negatively correlated with ORP and DO, indicating that the reducing environment under hypoxia is conducive to the survival and proliferation of sulfate-reducing bacteria (SRB), which facilitates the occurrence of bacterial sulfate reduction (BSR) processes [25]. The surface samples in April were significantly positively correlated with Chl *a* content, indicating that the oxygen–rich environment at the surface in April is conducive to the survival and proliferation of phototrophic bacteria.

By analyzing the Heatmap of community composition at the genus level (Appendix A), we observed that the compositions of the Jul-H and Oct-H samples are highly similar. Functional microorganisms such as the genus *Sulfuritalea* are typical chemoautotrophic bacteria capable of oxidizing thiosulfate (S_2_O_3_^2−^), elemental sulfur, or hydrogen (H_2_), as well as reducing nitrate [26]. *Methylomonas*, also known as methane-oxidizing bacteria, can utilize methyl-containing monocarbon compounds such as methane and methanol as both carbon and energy sources. The genus *Sulfurimonas*, belonging to the *Proteobacteria phylum*, is renowned for its ability to reduce nitrate, oxidize sulfur and hydrogen, and harbor Group IV hydrogenases [26,27]. Most members of the *Hydrogenophilaceae* family are microorganisms that employ various inorganic electron donors, such as reduced sulfur compounds or hydrogen, for chemolithotrophic or mixotrophic growth, and are involved in sulfide oxidation, methane oxidation, and nitrate reduction processes. The functional microorganisms driving the biogeochemical cycling of nutrients, as mentioned above, are abundant in hypoxic bottom waters during summer and autumn, whereas their relative abundance remains comparatively low in other categories (Appendix A). We generated a species composition heatmap by annotating with the *nosZ* gene of nitrous oxide (N_2_O) reductase as a marker gene (Appendix A). The heatmap primarily revealed species belonging to the phylum *Proteobacteria*, including genera such as *unclassified_o_Nitrosomonadales*, *Sulfuritalea*, *unclassified_f_Rhodocyclaceae*, and *Dechloromonas*. Furthermore, these species exhibited high relative abundance in hypoxic water bodies during summer and autumn, while their levels were extremely low in winter and spring. They serve as the primary drivers of the DNF process in water bodies, capable of ultimately converting NO_3_^−^ into N_2_ (Appendix A).

Heavy metal concentrations significantly affected the spatial differentiation of the planktonic bacterial community structure, with Mn, Co, Fe, Ni, As, and Cu contributing significantly to individual effects(Figure 7b). Among these, there was a negative correlation between heavy metals such as Mn, Co, Fe, Ni, As, and Cu, and these metals have opposite effects on species composition. Some heavy metals are essential elements involved in microbial cell synthesis, serving as important components of several key enzymes and playing a significant role in various redox reactions. For example, Cu is an important cofactor for several oxidative stress-related enzymes, including *catalase*, *superoxide dismutase*, *peroxidase*, *cytochrome c oxidase*, and ferrous oxidase [28]. Therefore, it is an essential nutrient that is incorporated into many metal enzymes involved in hemoglobin formation and carbohydrate metabolism. The ability of copper to cycle between its oxidized state (Cu(II)) and reduced state (Cu(I)) is utilized by copper enzymes involved in redox reactions. However, this characteristic of copper makes it potentially toxic, as the transformation between Cu(II) and Cu(I) can lead to the production of superoxides and hydroxyl radicals [26]. Nevertheless, high concentrations of heavy metals such as As and Cr may stress microbial growth [13,28]. Studies have also shown that the interaction between SRB and iron oxides can lead to the reduction and dissolution of iron oxides, as SRB undergo sulfate reduction to produce H_2_S, which reacts with iron oxides to cause their reduction and dissolution, thereby increasing the reducing iron content in a water body [19,29,30].

Hypoxia is a prerequisite for the enhancement of biogeochemical processes and their coupling in water bodies, including C, N, and S. It is also a key factor causing the increase in Fe, As, PO_4_^3−^, and DSi in water bodies. Therefore, monitoring real-time O_2_ concentration in water bodies can help understand the dynamic changes in redox conditions of water bodies, and predict the occurrence of C, N, and S-related biogeochemical processes in water bodies, providing data support for preventive measures in reservoir management.

## 5. Conclusions

(1)The As, Ni, Co, and Mn present in the water body of the study area are likely to predominantly originate from mine wastewater. The Zn, Pb, Cd, and Cr in the water body are primarily associated with domestic and agricultural sewage, as well as traffic emissions. Meanwhile, the Fe and Cu in the water body are sourced from natural origins.(2)Hypoxia serves as the most critical factor in expediting the cycling of nutrient salts and heavy metal elements in bottom water bodies. The O_2_ content plays a pivotal role in regulating processes such as sulfate reduction, nitrate reduction, Fe reduction, and arsenate reduction. The microbial-mediated biogeochemical cycling of elements, encompassing anaerobic decomposition of organic matter, sulfate reduction, nitrate reduction, sulfide oxidation, and anaerobic ammonium oxidation, influences the reductive dissolution of iron (oxy) hydroxides in suspended particulate matter and surface sediments, as well as the redox reactions of arsenic compounds. This, in turn, facilitates the accumulation of Fe, As, PO_4_^3−^, and DSi in the anoxic bottom waters and may elevate their concentrations throughout the water column during mixing periods, potentially leading to eutrophication and heavy metal contamination.(3)Therefore, monitoring the real-time O_2_ concentration in water bodies aids in understanding the dynamic variations in their redox conditions, thereby enabling the prediction of coupled nutrient and heavy metal processes, and providing data support for preventive measures in reservoir management. Additionally, the composition of microbial communities can reflect the redox conditions of water bodies, the biogeochemical cycling of elements, the rates of redox processes, as well as the occurrence and intensity of microbial-mediated Fe and As redox reactions. Periodic monitoring of changes in water body microbial communities can also effectively forecast trends in the biogeochemical processes of nutrients and heavy metals, and offer reasonable reservoir management strategies from a microbial ecology standpoint.

## Figures and Tables

**Figure 1 microorganisms-13-02750-f001:**
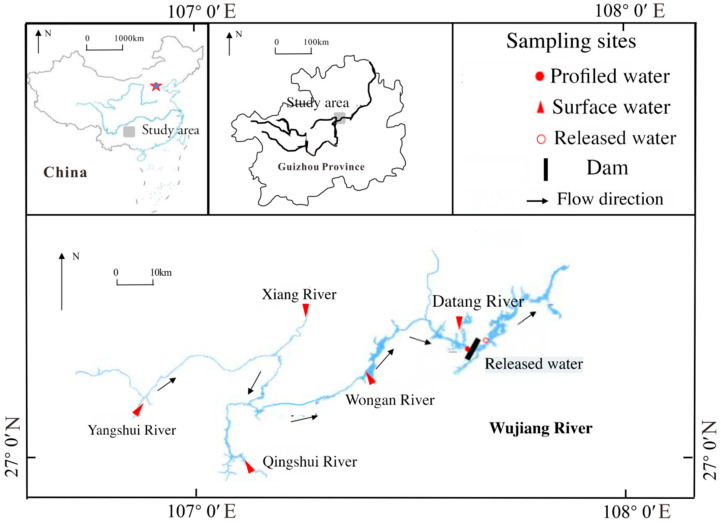
The sketch map of the study area and sampling sites.

**Figure 2 microorganisms-13-02750-f002:**
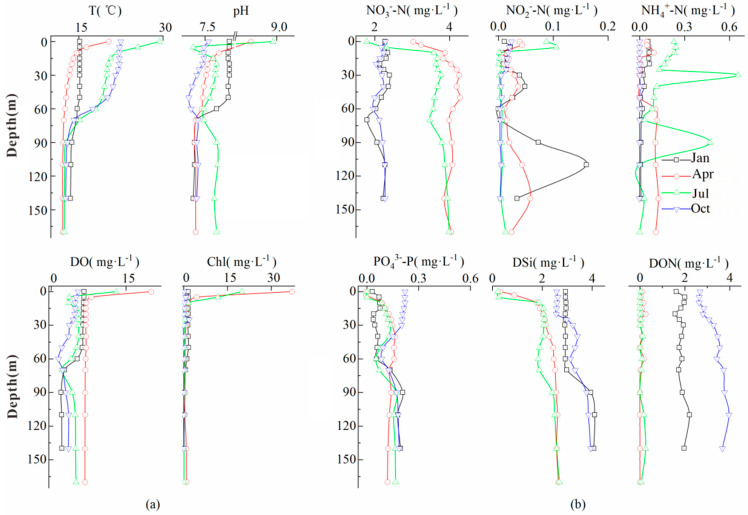
Temporal and spatial distribution of physico-chemical parameters (**a**) and nutrients (**b**) in the profile waters of the study area.

**Figure 3 microorganisms-13-02750-f003:**
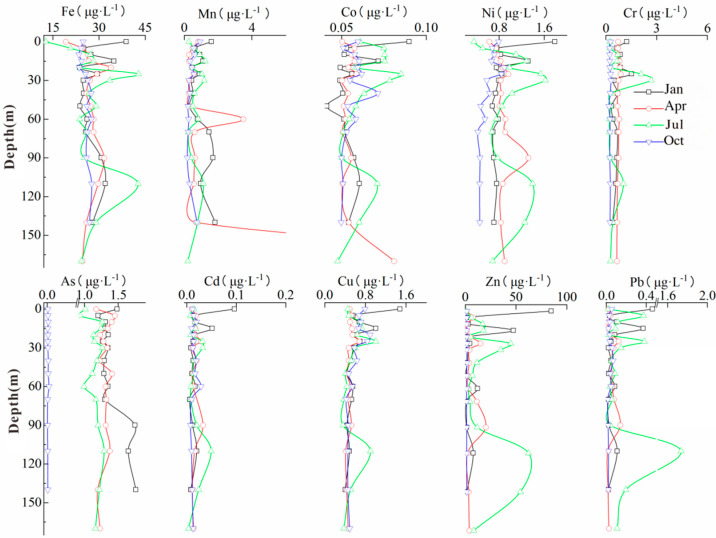
Temporal and spatial distribution of dissolved heavy metals in the study area.

**Figure 4 microorganisms-13-02750-f004:**
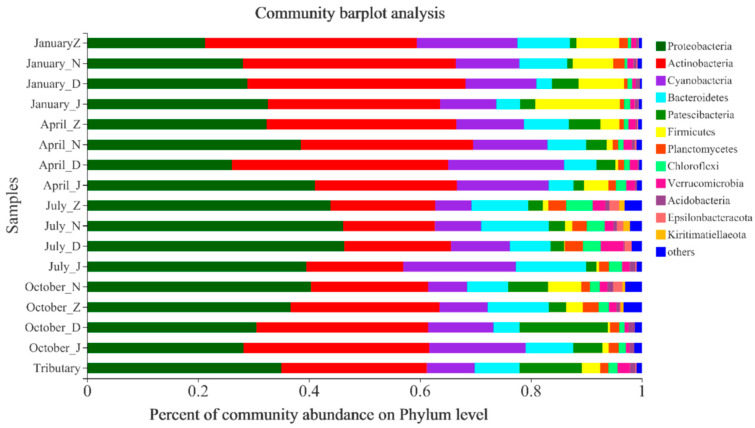
Seasonal differences in the composition of planktonic bacteria at the phylum level in different profiles.

**Figure 5 microorganisms-13-02750-f005:**
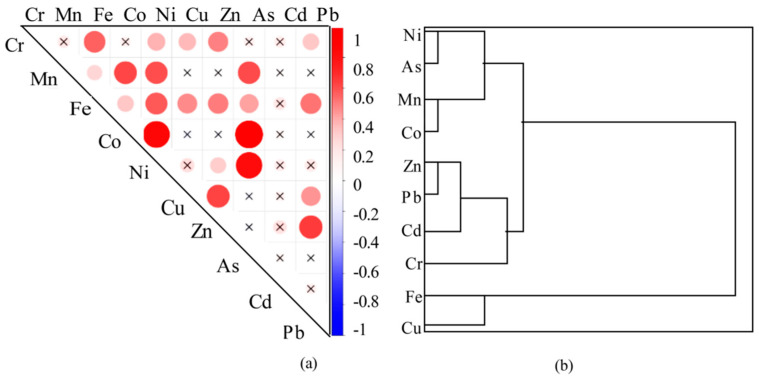
Correlation analysis and Cluster analysis among dissolved heavy metals: (**a**) visualization diagram of Pearson’s correlation coefficient, where × represents *p* > 0.05. The circle is darker and larger, and the correlation is stronger. (**b**) Cluster analysis (*n* = 84).

**Figure 6 microorganisms-13-02750-f006:**
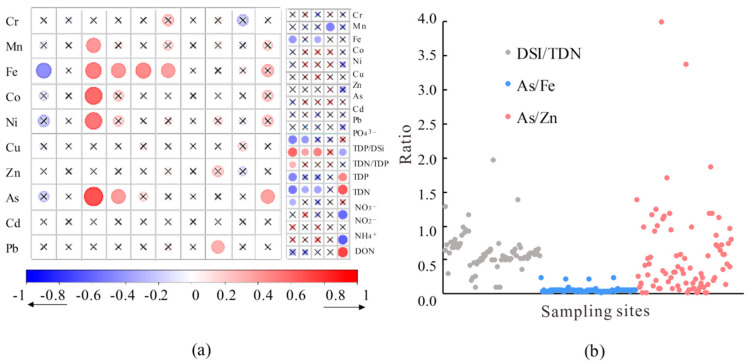
Correlation analysis among different factors: (**a**) visualization diagram of Pearson’s correlation coefficient between nutrients, water chemical parameters, and dissolved heavy metals, × represents *p* > 0.05, *n* = 84. (**b**) The scatter plots of characteristic ratios of the heavy metals and nutrients.

**Figure 7 microorganisms-13-02750-f007:**
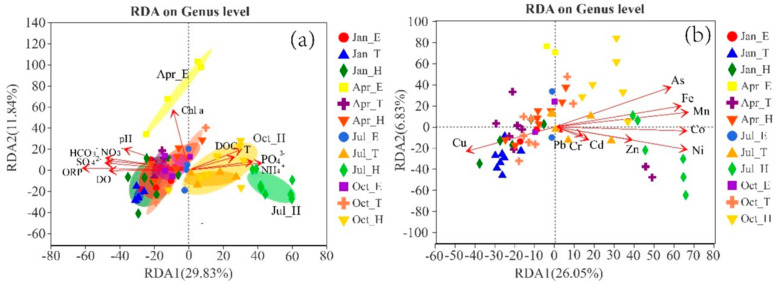
RDA analysis of the effect of water chemistry (**a**) and heavy metal concentration (**b**) on the composition of the bacterioplankton community at the genus level.

## Data Availability

The original contributions presented in this study are included in the article/Appendix A. Further inquiries can be directed to the corresponding author.

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
