# Peer review of "Study on the Source and Microbial Mechanisms Influencing Heavy Metals and Nutrients in a Subtropical Deep-Water Reservoir"

_microorganisms, 2025, doi:10.3390/microorganisms13122750_

Round 1
Reviewer 1 Report
Comments and Suggestions for Authors
The article is devoted to the microbial mechanisms driving the interactions between heavy metals and nutrients in water bodies using the example of The Gupitan Reservoir. The temporal and spatial variations in heavy metal and nutrients and the metagenomic composition of the reservoir water were analyzed in January, April, July and October 2019. The authors found that the sources of As, Ni, Co and Mn were mining wastewater, while Zn, Pb, Cd and Cr were associated with the discharge of domestic and agricultural wastewater. In the studied area, Proteobacteria, Actinobacteria, Cyanobacteria and Bacteroidetes dominate, with the share of dominant types reaching 90%. The parameters dissolved oxygen (DO) concentration and pH strongly influence the structure of the microbial community at the bottom of the reservoir. The relationship between heavy metals and nutrients in the reservoir of the study area is primarily influenced by a combination of factors, including runoff, sulfate reduction, and adsorption and desorption of particulate pollutants.
The following methods were used to conduct the study: Water temperature, DO, pH, total dissolved solids (TDS), and conductivity were measured in situ. The concentrations of nutrients in the water were measured using a Skalar San++ continuous flow analyzer from the Netherlands. The concentrations of heavy metals were determined by inductively coupled plasma mass spectrometry. Metagenomic sequencing was conducted using the Illumina HiSeq Xten sequencing platform from the United States.
The manuscript is well-structured, and is in the journal's field.
The results obtained are illustrated in 7 figures.
The number of sources cited is 25, of which 11 are in the period 2020-2025. An article appears twice in a references. The number of self-citations is 6, with the cited publications being relevant to the topic of the study.
The proposed experimental design is suitable for studying the mechanisms of heavy metal transformations and the role of microbial communities in seasonally stratified lakes and reservoirs.
The results obtained are of great interest to authors working in the same scientific field.
The article contains new original data and after corrections is suitable for publication in the journal Microorganisms.
Questions and comments
Materials and Methods
Line 82: “The total water resources in the basin are 10.4259 million kW, of which 5.804 million kW are in the main stream of the Wujiang River”
The statement "total water resources in the basin are million kW" is inaccurate because kilowatt (kW) is a unit of power, not a unit of water volume. To correctly describe water resources, you should use units of volume, such as cubic meters (m³) or cubic kilometers (km³)
Line 114: The concentrations of nutrients (TDN, NH4+, NO3–, NO2–, TDP, PO43– and DSi) in the water were measured using a Skalar San++ continuous flow analyzer from the Netherlands
It is necessary to describe the abbreviations TDN, TDP and DSi, as they appear in the text for the first time.
Materials and methods lack information on the method used to determine the chlorophyll a (Chl a) content.
The Results section comments on the impact of the ORP on the other indicators, but it is not listed in Materials and Methods.
- Results
Line 168: „…order: Fe > Zn > Mn, As, Ni > Cu, Cr > Pb > Co > Cd (Table 2).“
Table 2 is Table S3 presented in the supplementary materials:
Table S3 Concentration ranges of dissolved heavy metals in the study
In “Materials and methods” it is mentioned about surface water which was taken from the tributaries (Yangshui River, Xiangjiang River, Datang River, Wengan River, and Qingshui River),) and discharged water bodies. It is necessary to present sufficient information on physicochemical and chemical parameters of these samples either in the article or in the supplementary materials.
- Discussion
Lines 211 and 212: “This has also been confirmed in the Weng'an River and Qingshui River (Figure 3).”
Figure 3 provides information about temporal and spatial distribution of dissolved heavy metals in the profile water only in the Aha reservoir.
Lines 227-235: “ORP(Oxidation-Reduction Potential) was significantly positively correlated with TDP, DSi, and DON concentrations, whereas DO was significantly negatively correlated with TDP, DSi, and DON concentrations (Figure 5a)…..”.
In Figure 5a, the parameters ORP, DO and Chl a are missing.
Lines 288- 292: “In July and October, the bottom samples were negatively correlated with ORP, DO, and SO42−, indicating that the reducing environment under hypoxia is conducive to the survival and proliferation of sulfate-reducing bacteria (SRB), which facilitates the occurrence of bacterial sulfate reduction (BSR) processes“.
There is no information anywhere about measuring the concentration of sulfates.
Deltaproteobacteria, to which sulfate-reducing bacteria belong, are missing from Figure 6, and their presence is evidence for the above statement.
For a more accurate presentation of the information, it is necessary to renumber the figures (figure 6 to become figure 4, figure 4 to become figure 5, and figure 5 to become figure 6).
- Conclusions
Lines 322-325: „Combining the DSi/TDN, As/Fe, and Zn/As ratios, the coupling between heavy metals and nutrients in the water body of the study area is influenced primarily by a combination of factors, including runoff, sulfate reduction, and the adsorption and desorption of particulate pollutants“.
Ferric reduction and manganese (IV) reduction processes are also involved. This can be seen in the figure 3.
References
N 13: Zhang, X.; Ding, S.; Lv, H.; Cui, G.; Yang, M.; Wang, Y.; Guan, T.; Li, X. Microbial controls on heavy metals and nutrients simultaneous release in a seasonally stratified reservoir. Environ Sci Pollut Res 2022, 29, 1937-1948.
N 24 Zhang, X.; Ding, S.; Lv, H.; Cui, G.; Yang, M.; Wang, Y.; Guan, T.; Li, X. Microbial controls on heavy metals and nutrients simultaneous release in a seasonally stratified reservoir. Environ Sci Pollut R. 2021, 1-12.
The same publication is cited under two numbers.
Recommendation
A huge amount of work has been done. The authors have information from metagenomic analyzes that allows a figure to be prepared with the relative abundance (%) of the bacterial genera or family. This information would provide more knowledge about the functional relationship of bacteria carrying out various processes in the studied microbial community and their influence on the behavior of heavy metals and nutrients.
Figure 7 is not informative enough: Figure 7. RDA analysis of the effect of water chemistry (a) and heavy metal concentration (b) on the composition of the bacterioplankton community at the genus level.
Author Response
Comments 1: Line 82: “The total water resources in the basin are 10.4259 million kW, of which 5.804 million kW are in the main stream of the Wujiang River”. The statement "total water resources in the basin are million kW" is inaccurate because kilowatt (kW) is a unit of power, not a unit of water volume. To correctly describe water resources, you should use units of volume, such as cubic meters (m³) or cubic kilometers (km³)
Response 1: Thank you for your comments. As suggested, we have changed the “water resource” to “power generation capacity” (Lines 85-86). In addition, the water resource (Expressed in cubic meters and cubic kilometers) of Goupitan Reservoir are also described in detail in the following text (Lines 90-92).
Comments 2: Line 114: The concentrations of nutrients (TDN, NH4+, NO3–, NO2–, TDP, PO43– and DSi) in the water were measured using a Skalar San++ continuous flow analyzer from the Netherlands. It is necessary to describe the abbreviations TDN, TDP and DSi, as they appear in the text for the first time.
Response 2: Thank you for your comments. As suggested, we have completed the full names of TDN, TDP, and DSi in the revised manuscript (Lines 113-114).
Comments 3: Materials and methods lack information on the method used to determine the chlorophyll a (Chl a) content.
Response 3: Thank you for your comments. As suggested, we have added the testing method for chlorophyll a in the revised manuscript (Lines 113-114).
Comments 4: The Results section comments on the impact of the ORP on the other indicators, but it is not listed in Materials and Methods.
Response 4: Thank you for your comments. As suggested, we have added the testing method for oxidation-reduction potential (ORP) in the revised manuscript (Lines 113-114).
Comments 5: Line 168: …order: Fe > Zn > Mn, As, Ni > Cu, Cr > Pb > Co > Cd (Table 2). Table 2 is Table S3 presented in the supplementary materials: Table S3 Concentration ranges of dissolved heavy metals in the study.
Response 5: Thank you for your comments. As suggested, we have made corresponding modifications in the revised manuscript (Line 187).
Comments 6: In “Materials and methods” it is mentioned about surface water which was taken from the tributaries (Yangshui River, Xiangjiang River, Datang River, Wengan River, and Qingshui River) and discharged water bodies. It is necessary to present sufficient information on physicochemical and chemical parameters of these samples either in the article or in the supplementary materials.
Response 6: Thank you for your comments. As suggested, we have added the physical and chemical parameter values of all tributaries in the supplementary materials (Table S4).
Comments 7: Lines 211 and 212: “This has also been confirmed in the Weng'an River and Qingshui River (Figure 3).” Figure 3 provides information about temporal and spatial distribution of dissolved heavy metals in the profile water only in the Aha reservoir.
Response 7: Thank you for your comments. There is indeed no relevant information reflected in Figure 3; we have made corresponding corrections in the revised manuscript, changing Figure 3 to Table S4.
Comments 8: Lines 227-235: “ORP(Oxidation-Reduction Potential) was significantly positively correlated with TDP, DSi, and DON concentrations, whereas DO was significantly negatively correlated with TDP, DSi, and DON concentrations (Figure 5a)…..”. In Figure 5a, the parameters ORP, DO and Chl a are missing.
Response 8: Thank you for your comments. We actually conducted a correlation analysis between all parameters, but only a portion was extracted and shown in Figure 5. Redundancy analysis also includes relevant information. Therefore, we have indicated significance and Figure 7 in the revised manuscript, and detailed explanations were provided (Lines 247-251).
Comments 9: Lines 288- 292: “In July and October, the bottom samples were negatively correlated with ORP, DO, and SO42−, indicating that the reducing environment under hypoxia is conducive to the survival and proliferation of sulfate-reducing bacteria (SRB), which facilitates the occurrence of bacterial sulfate reduction (BSR) processes“. There is no information anywhere about measuring the concentration of sulfates. Deltaproteobacteria, to which sulfate-reducing bacteria belong, are missing from Figure 6, and their presence is evidence for the above statement.
Response 9: Thank you for your comments. We have added a heatmap consisting of SRMs/SOMs annotated with aprAB and dsrAB in the attachment (Figure S2), and a detailed explanation (As shown in Figure S2, Sulfuritalia, unclassifiedd_f_Thiobacillaceae, unclassifiedd_o'Nitrosomonaales belong to Nitrosomonaales (order Methylomonas), which are a class of chemoautotrophic microorganisms capable of oxidizing sulfides, thiosulfates, and elemental sulfur and reducing nitrates) was provided in the revised manuscript (Lines 315-320).
Comments 10: For a more accurate presentation of the information, it is necessary to renumber the figures (figure 6 to become figure 4, figure 4 to become figure 5, and figure 5 to become figure 6).
Response 10: Thank you for your comments. As suggested, we have renumbered the images and made corresponding modifications in the revised manuscript.
Comments 11: Lines 322-325: “Combining the DSi/TDN, As/Fe, and Zn/As ratios, the coupling between heavy metals and nutrients in the water body of the study area is influenced primarily by a combination of factors, including runoff, sulfate reduction, and the adsorption and desorption of particulate pollutants”. Ferric reduction and manganese (IV) reduction processes are also involved. This can be seen in the figure 3.
Response 10: Thank you for your comments. We have made corresponding modifications in the revised manuscript (Line 364).
Comments 12: N 13: Zhang, X.; Ding, S.; Lv, H.; Cui, G.; Yang, M.; Wang, Y.; Guan, T.; Li, X. Microbial controls on heavy metals and nutrients simultaneous release in a seasonally stratified reservoir. Environ Sci Pollut Res 2022, 29, 1937-1948.
N 24 Zhang, X.; Ding, S.; Lv, H.; Cui, G.; Yang, M.; Wang, Y.; Guan, T.; Li, X. Microbial controls on heavy metals and nutrients simultaneous release in a seasonally stratified reservoir. Environ Sci Pollut R. 2021, 1-12.
The same publication is cited under two numbers.
Response 10: Thank you for your comments. We have removed duplicate references and changed the corresponding reference numbers in the revised manuscript.
Comments 13: A huge amount of work has been done. The authors have information from metagenomic analyzes that allows a figure to be prepared with the relative abundance (%) of the bacterial genera or family. This information would provide more knowledge about the functional relationship of bacteria carrying out various processes in the studied microbial community and their influence on the behavior of heavy metals and nutrients.
Figure 7 is not informative enough: Figure 7. RDA analysis of the effect of water chemistry (a) and heavy metal concentration (b) on the composition of the bacterioplankton community at the genus level.
Response 10: Thank you for your comments. We have added more information about the physicochemical factors and metagenome in the study area in the supplementary materials (Figure S2 and Table S4).
Reviewer 2 Report
Comments and Suggestions for Authors
- General comments: The investigation in the manuscript is fine. However, it has major deficiencies in the presntation of information. The manuscript needs proof reading and in precision in snnetence constructions. Please find step-by-step reviewer comments to each of the presneted information. The manuscript seems to have benn submitted without proof reading the accuracy of the title. Additionally, there are a number of instances where the methods are not described, yet authors mention them.
- Abstract; The abstract casually mentioned micronbial mechanisms without pinpointing the specific mechanisms probed or proposed sources f the heavy metals.
- Introduction section: The Introduction section suffers from lack of coherence in presntation. The ideas presnted are not developed or developed from one step to the next. Ideas such as eutrophicatio, metal speciation andother ideas are presented without being pursued in the manuscfipt. Introduction section is therefore poorly presnted in terms of coherence. Major paragraphs are presented as dicrete separate ideas. The last paragraph in the introduction section (Lines 64 - 73) mention "mechanisms of coupling metal concnetrations and neutrient salts (lines 66 -68)" These are simply coorelations presented as mechanisms. Specific mechanisms need to be presnted in the Introduction section and should be distinguished from correlations pursued. It is also nted that authors use the term " Theoretical basis" (line 72). Why is this a theretical basis yet, experimental data are statiscailly treated , which lead to conclusons?
- Material and Methods: (a) Please citae relevant references at the end of paragraph in lines 77 - 83. (b) Study Area: This section can be improved. using a better summary in linses 76 -95. The latitudes and longitudes of the study area will be preferable. (c) Section 2.2: Please use the word analysis instead of analyzing in line 97. It is also recommended that authors be ocnsistent in word usage: For examaple int he picture in Figure 1; is it Xiang or Xiangjiang (line 100). Line 100 and Figure 1 naming label should match.
- in Sdction 2.2; Given that samples were acidified with nitric acid in the filed, how then were the NH4+-N, nitrates, nitrites dtermined given the acification procdure done. Such information must be made clear, without which the dta may be deemed unreliable/invalid.
- Please provie the website for the softwae used in line 129. In addition, please exaplin in full words before using an aabbrevaiton what the letters KO vlues stand for.
- Results section: UThere is no mention on how the chrolophyll A concnetration was determined in section 3.1. Such information is to be presented in the Procedures section.
- Details on the ICP-MS procedures are not presnted in section 2.2. Such information is crucial to the aunthetication/validation of the presnted data. What were the LODs , LOQs found?
- Overall: This manuscripts needs reorganization in the procedures section, validation of methods before presentation of the examined data. This is a mjaor wekaness. In view of this, this reviewer is reluactantt in recommneding the ms for publication in its presnt form. Mjaor revision is needed in order to meet validatin of data, and the accuracy of the metods aand data found.
- The Figures are clelay presented.
- In conclusion: major corrections, coherent presention of data, and care must be exercised in the revision.
- General comments: The investigation in the manuscript is fine. However, it has major deficiencies in the presntation of information. The manuscript needs proof reading and in precision in snnetence constructions. Please find step-by-step reviewer comments to each of the presneted information. The manuscript seems to have benn submitted without proof reading the accuracy of the title. Additionally, there are a number of instances where the methods are not described, yet authors mention them.
- Abstract; The abstract casually mentioned micronbial mechanisms without pinpointing the specific mechanisms probed or proposed sources f the heavy metals.
- Introduction section: The Introduction section suffers from lack of coherence in presntation. The ideas presnted are not developed or developed from one step to the next. Ideas such as eutrophicatio, metal speciation andother ideas are presented without being pursued in the manuscfipt. Introduction section is therefore poorly presnted in terms of coherence. Major paragraphs are presented as dicrete separate ideas. The last paragraph in the introduction section (Lines 64 - 73) mention "mechanisms of coupling metal concnetrations and neutrient salts (lines 66 -68)" These are simply coorelations presented as mechanisms. Specific mechanisms need to be presnted in the Introduction section and should be distinguished from correlations pursued. It is also nted that authors use the term " Theoretical basis" (line 72). Why is this a theretical basis yet, experimental data are statiscailly treated , which lead to conclusons?
- Material and Methods: (a) Please citae relevant references at the end of paragraph in lines 77 - 83. (b) Study Area: This section can be improved. using a better summary in linses 76 -95. The latitudes and longitudes of the study area will be preferable. (c) Section 2.2: Please use the word analysis instead of analyzing in line 97. It is also recommended that authors be ocnsistent in word usage: For examaple int he picture in Figure 1; is it Xiang or Xiangjiang (line 100). Line 100 and Figure 1 naming label should match.
- in Sdction 2.2; Given that samples were acidified with nitric acid in the filed, how then were the NH4+-N, nitrates, nitrites dtermined given the acification procdure done. Such information must be made clear, without which the dta may be deemed unreliable/invalid.
- Please provie the website for the softwae used in line 129. In addition, please exaplin in full words before using an aabbrevaiton what the letters KO vlues stand for.
- Results section: UThere is no mention on how the chrolophyll A concnetration was determined in section 3.1. Such information is to be presented in the Procedures section.
- Details on the ICP-MS procedures are not presnted in section 2.2. Such information is crucial to the aunthetication/validation of the presnted data. What were the LODs , LOQs found?
- Overall: This manuscripts needs reorganization in the procedures section, validation of methods before presentation of the examined data. This is a mjaor wekaness. In view of this, this reviewer is reluactantt in recommneding the ms for publication in its presnt form. Mjaor revision is needed in order to meet validatin of data, and the accuracy of the metods and data found.
- The Figures are clelay presented.
- In conclusion: major corrections, coherent presention of data, and care must be exercised in the revision.
Author Response
Comment 1: General comments: The investigation in the manuscript is fine. However, it has major deficiencies in the presntation of information. The manuscript needs proof reading and in precision in snnetence constructions. Please find step-by-step reviewer comments to each of the presneted information. The manuscript seems to have benn submitted without proof reading the accuracy of the title. Additionally, there are a number of instances where the methods are not described, yet authors mention them.
Response 1: Thank you for your comments. We have made revisions to each of the above issues in the revised draft, and highlighted the modified parts in red.
Comment 2: Abstract; The abstract casually mentioned micronbial mechanisms without pinpointing the specific mechanisms probed or proposed sources f the heavy metals.
Response 2: Thank you for your comments. We have made modifications to the above issues in the revised manuscript (Lines 15-20).
Comment 3: Introduction section: The Introduction section suffers from lack of coherence in presntation. The ideas presnted are not developed or developed from one step to the next. Ideas such as eutrophicatio, metal speciation andother ideas are presented without being pursued in the manuscfipt. Introduction section is therefore poorly presnted in terms of coherence. Major paragraphs are presented as dicrete separate ideas. The last paragraph in the introduction section (Lines 64 - 73) mention "mechanisms of coupling metal concnetrations and neutrient salts (lines 66 -68)" These are simply coorelations presented as mechanisms. Specific mechanisms need to be presnted in the Introduction section and should be distinguished from correlations pursued. It is also nted that authors use the term " Theoretical basis" (line 72). Why is this a theretical basis yet, experimental data are statiscailly treated , which lead to conclusons?
Response 3: Thank you for your comments. We have improved the introduction section in the revised manuscript and deleted and modified inappropriate statements
Comment 4: Material and Methods: (a) Please citae relevant references at the end of paragraph in lines 77 - 83. (b) Study Area: This section can be improved. using a better summary in linses 76 -95. The latitudes and longitudes of the study area will be preferable. (c) Section 2.2: Please use the word analysis instead of analyzing in line 97. It is also recommended that authors be ocnsistent in word usage: For examaple int he picture in Figure 1; is it Xiang or Xiangjiang (line 100). Line 100 and Figure 1 naming label should match.
Response 4: Thank you for your comments.We have improved the introduction materials and methods section in the revised manuscript (Lines 86-87, lines 113-139).
Comment 5: in Sdction 2.2; Given that samples were acidified with nitric acid in the filed, how then were the NH4+-N, nitrates, nitrites dtermined given the acification procdure done. Such information must be made clear, without which the dta may be deemed unreliable/invalid.
Response 5: Thank you for your comments. We have provided detailed supplements to the methodology section in the revised manuscript, including pre-processing, sampling process, blank sample setting, error range, etc (lines 125-136, lines 150-152).
Comment 6: Please provie the website for the softwae used in line 129. In addition, please exaplin in full words before using an aabbrevaiton what the letters KO vlues stand for.
Response 6: Thank you for your comments. We have explained the kO value in the revised manuscript and provided explanations for all software (Lines 148-153).
Comment 7: Results section: UThere is no mention on how the chrolophyll A concnetration was determined in section 3.1. Such information is to be presented in the Procedures section.
Response 7: Thank you for your comments. We have supplemented the measurement method of chlorophyll in the revised manuscript (lines 113-114).
Comment 8: Details on the ICP-MS procedures are not presnted in section 2.2. Such information is crucial to the aunthetication/validation of the presnted data. What were the LODs , LOQs found?
Response 8: Thank you for your comments. We have supplemented the measurement methods for all indicators in the revised manuscript (lines 125-136).In addition, ICP-MS is not a website or software, but an Inductively Coupled Plasma Mass Spectrometer.
Comment 9: Overall: This manuscripts needs reorganization in the procedures section, validation of methods before presentation of the examined data. This is a mjaor wekaness. In view of this, this reviewer is reluactantt in recommneding the ms for publication in its presnt form. Mjaor revision is needed in order to meet validatin of data, and the accuracy of the metods and data found.
Response 9: Thank you for your comments.We have added more raw data and supplementary analysis in the attachment, hoping to meet the requirements of the reviewers. In addition, we have made significant revisions to the logic and wording of the entire text.
Comment 10: The Figures are clelay presented.
Response 10: Thank you for your comments. We have optimized the resolution of the images and added Figure S2 in the attachment to supplement the macro gene information involved in the study area
Comment 11: In conclusion: major corrections, coherent presention of data, and care must be exercised in the revision.
Response 11: Thank you for your comments. We have made significant revisions to the conclusion section, highlighting key points and removing unnecessary parts (Lines 357-372).
Reviewer 3 Report
Comments and Suggestions for Authors
The manuscript investigates heavy‑metal sources and microbial processes in a subtropical deep‑water reservoir through seasonal water sampling (January, April, July, and October 2019), analyzes nutrient and heavy‑metal distributions, and assesses planktonic bacterial communities using metagenomic approaches; however, the current presentation exhibits numerous issues requiring major revision. The manuscript suffers from serious issues with the English language, unclear methodology, poor figure/table quality, missing QA/QC, and limited novelty. Recommendation: Major Revision.
The title is grammatically incomplete. The phrase “Study on the of Source and Microbial Mechanisms” is nonsensical, and the enumeration of “Source and Microbial Mechanisms” does not clearly define the scope or variables studied. A concise and descriptive title should be provided.
The abstract partly describes reservoir selection, sampling seasons, and major findings, yet it contains run‑on sentences and lacks clarity. The abstract should state the objectives, methods (number of samples, analytical techniques), key numerical results with confidence intervals, and the implications of the findings.
The keywords provided are not optimal; they duplicate words from the title.
Introduction
The introduction cites some literature on eutrophication and heavy metals, yet recent global studies on metal–nutrient coupling and microbial interactions are sparsely discussed. Many references are more than a decade old; for example, classic works from 2006 and 2013 are cited, whereas recent high‑impact studies from 2024–2025 are missing. The discussion also lacks critical comparison with similar reservoirs worldwide; sources of heavy metals are described without referencing established geochemical models. The authors should review current literature and incorporate recent findings.
Authors should clarify how their dataset adds beyond previous reservoir studies.
Material and methods
Improve the presentation of Figure 1.
The methods mention that relative deviations were < 5% and that standard curves had correlation coefficients > 99.9%; however, no details of blanks, standard reference materials, calibration procedures, or duplicate analyses are provided. Sample preservation, acidification, and holding times should be described with reference to recognized standards. Without clear QA/QC protocols, the reliability of the analytical results cannot be assessed.
Sampling: The authors should provide exact coordinates of sampling sites, depths sampled, and replicate numbers. Were samples collected in triplicate? Were contamination controls applied?
Instruments are cited, but key operating parameters (e.g., detection limits, operating conditions, calibration standards) are absent. The ICP‑MS method should state the isotopes measured, internal standard concentration, and interference correction. For nutrient analyses, details such as method detection limits for NO₃⁻, NH₄⁺, etc., should be provided. Without such details, the method cannot be reproduced.
The authors mention using DIAMOND v0.8.35 for sequence alignment, but the statistical analyses (Pearson correlation, cluster analysis, and redundancy analysis) lack any software or package description. Provide software names and versions.
Results and discussion
Although the study focuses on Goupitan Reservoir in China, the authors claim to provide “guidance for addressing combined pollution issues in karst deep reservoirs”. However, the data interpretation is largely site‑specific, with limited comparison to global reservoirs. To enhance broader relevance, results should be contextualized with similar studies worldwide and generalized mechanisms rather than site‑specific observations.
Coverage of critical analytes – Only a subset of heavy metals (Fe, Zn, Mn, As, Ni, Cu, Cr, Pb, Co, Cd) was measured. Mercury (Hg), one of the most toxic and commonly regulated metals, was omitted, as were key metalloids such as selenium and antimony. Nutrient analyses do not include total phosphorus or chlorophyll b. Given the reservoir’s location near phosphorus mines, arsenic speciation and methylated forms could provide deeper insights. Include all relevant contaminants or explain omissions.
A table comparing results with other subtropical or global reservoirs would contextualize findings.
Statements such as “As, Ni, Co, and Mn were derived primarily from mine wastewater, whereas Zn, Pb, Cd, and Cr were related to domestic and agricultural wastewater” rely solely on correlations without source apportionment modelling. Similarly, the claim that heavy metals significantly affected the spatial differentiation of bacterioplankton requires multivariate analysis to separate effects from confounding variables (depth, season). The conclusion that runoff input played a significant role is not supported by quantitative runoff data. The manuscript should avoid over‑interpretation and provide evidence for causal claims.
Contradiction arises between the assertion that “elemental Cd does not exhibit a clear pattern with depth” and the earlier claim that Cd is associated with agricultural wastewater. These conflicting interpretations should be reconciled with a more rigorous analysis.
The conclusion restates the results but does not synthesize the findings. It reiterates sources of metals and microbial dominance without linking back to hypotheses or discussing management implications. The conclusion should succinctly summarize major findings, acknowledge limitations, and suggest future research directions.
References: Some cited literature appears tangential to the study. For example, the reference on “Accurate prediction of soil heavy metal pollution using an improved machine learning method” pertains to machine‑learning prediction in soils and has little relevance to reservoir water studies. Similarly, several soil‑oriented papers (references [6], [7], [24], [26]) deal with terrestrial systems rather than aquatic reservoirs. Removing these and replacing them with studies on heavy‑metal dynamics in reservoirs or lakes would improve the literature review.
Author Response
Comment 1: The title is grammatically incomplete. The phrase “Study on the of Source and Microbial Mechanisms” is nonsensical, and the enumeration of “Source and Microbial Mechanisms” does not clearly define the scope or variables studied. A concise and descriptive title should be provided.
Response 1: Thank you for your comments. We will change the article title to: Impact of Heavy Metals and Nutrients on Microbial Community Composition in a Subtropical Deep-Water Reservoir. Besides, we have added more information on the physicochemical factors and metagenome that affect the concentration and distribution of heavy metals in the study area in the revised manuscript and attachments (Figure S2 and Table S4). Therefore, the current title is more in line with the main body of the article and is also consistent with the scientific questions raised in the abstract.
Comments 2: The abstract partly describes reservoir selection, sampling seasons, and major findings, yet it contains run‑on sentences and lacks clarity. The abstract should state the objectives, methods (number of samples, analytical techniques), key numerical results with confidence intervals, and the implications of the findings.
Response 2: Thank you for your comments. We have made revisions to the abstract in the revised manuscript, highlighting the research focus, clarifying the research background and significance, and simplifying the research conclusions (Lines 14-20).
Comments 3: The keywords provided are not optimal; they duplicate words from the title.
Response 3: Thank you for your comments. We have modified the keywords in the revised manuscript.
Comments 4: The introduction cites some literature on eutrophication and heavy metals, yet recent global studies on metal–nutrient coupling and microbial interactions are sparsely discussed. Many references are more than a decade old; for example, classic works from 2006 and 2013 are cited, whereas recent high‑impact studies from 2024-2025 are missing. The discussion also lacks critical comparison with similar reservoirs worldwide; sources of heavy metals are described without referencing established geochemical models. The authors should review current literature and incorporate recent findings. Authors should clarify how their dataset adds beyond previous reservoir studies.
Response 4: Thank you for your comments. More than half of the references in the introduction are from the past three years, and more than 80% are from the past five years. Therefore, our research status is quite appropriate. In addition, we focus on the coupling mechanism of nutrients, heavy metals, and microorganisms in a single deep-water reservoir within the region, and pay less attention to the global concentration of heavy metals and geochemical models in reservoirs. This is not the focus of this study, and we will add corresponding content in future research.
Comments 5: Improve the presentation of Figure 1.
Response 5: Thank you for your comments. We have indicated the location of the sampling points (including the division of surface water and profile water), the study area and its affiliated areas, latitude and longitude, and other information in Figure 1. Therefore, we believe that the current sampling map is sufficient to meet the needs of this study.
Comments 6: The methods mention that relative deviations were < 5% and that standard curves had correlation coefficients > 99.9%; however, no details of blanks, standard reference materials, calibration procedures, or duplicate analyses are provided. Sample preservation, acidification, and holding times should be described with reference to recognized standards. Without clear QA/QC protocols, the reliability of the analytical results cannot be assessed.
Response 6: Thank you for your comments. We have added information on blank samples, standard substances, analytical accuracy, parallel samples, and sample preservation, acidification, etc. during the experimental process in the revised manuscript (Lines 124-129, lines 133-137).
Comments 7: Sampling: The authors should provide exact coordinates of sampling sites, depths sampled, and replicate numbers. Were samples collected in triplicate? Were contamination controls applied?
Response 7: Thank you for your comments. We have added the latitude and longitude of the water samples from the reservoir area and the discharge water in the revised manuscript (lines 105-107). Secondly, the sampling map has clearly marked the scope and latitude and longitude range of the study area. We believe that this has provided sufficient information about the location of the sampling points. We have also explained the pollution control and parallel samples (Lines 124-129, lines 133-137).
Comments 8: Instruments are cited, but key operating parameters (e.g., detection limits, operating conditions, calibration standards) are absent. The ICP‑MS method should state the isotopes measured, internal standard concentration, and interference correction. For nutrient analyses, details such as method detection limits for NH4+, NO3–, etc., should be provided. Without such details, the method cannot be reproduced.
Response 8: Thank you for your comments. We have added detection limits, operating procedures, calibration ranges, as well as internal standard concentrations and interference correction for ICP-MS in the revised manuscript (Lines 113-136).
Comments 9: The authors mention using DIAMOND v0.8.35 for sequence alignment, but the statistical analyses (Pearson correlation, cluster analysis, and redundancy analysis) lack any software or package description. Provide software names and versions.
Response 9: Thank you for your comments. We have added software and methods related to statistical analysis in the revised manuscript (Lines 149-152).
Comments 10: Although the study focuses on Goupitan Reservoir in China, the authors claim to provide “guidance for addressing combined pollution issues in karst deep reservoirs”. However, the data interpretation is largely site‑specific, with limited comparison to global reservoirs. To enhance broader relevance, results should be contextualized with similar studies worldwide and generalized mechanisms rather than site‑specific observations.
Response 10: Thank you for your comments. Our research is a regional study, focusing on deep-water reservoirs in karst plateaus. We attempt to elucidate the effects of heavy metal and nutrient coupling on microbial communities and structures in reservoirs, but do not involve large-scale research. Therefore, our description of the significance of the research is cautious. Thank you for your understanding.
Comments 11: Coverage of critical analytes - Only a subset of heavy metals (Fe, Zn, Mn, As, Ni, Cu, Cr, Pb, Co, Cd) was measured. Mercury (Hg), one of the most toxic and commonly regulated metals, was omitted, as were key metalloids such as selenium and antimony. Nutrient analyses do not include total phosphorus or chlorophyll b. Given the reservoir’s location near phosphorus mines, arsenic speciation and methylated forms could provide deeper insights. Include all relevant contaminants or explain omissions.
Response 11: Thank you for your comments. Regrettably, our research indicators do not involve mercury, total phosphorus, or chlorophyll b. We fully agree with the reviewer's opinion and will add the determination of these indicators in future studies.
Comments 12: A table comparing results with other subtropical or global reservoirs would contextualize findings.
Response 12: Thank you for your comments. As mentioned earlier, our research focuses on the spatiotemporal distribution and coupling mechanism of nutrients and heavy metals in a single reservoir, as well as their impact on microbial communities and distribution. This manuscript is a regional study, and the geological background and socio-economic conditions of reservoirs in different regions vary greatly, resulting in significant spatiotemporal differences in nutrients, heavy metals, and microbial communities. Therefore, we did not involve a comparison of global reservoirs in our research. In the future, our research may involve related fields.
Comments 13: Statements such as “As, Ni, Co, and Mn were derived primarily from mine wastewater, whereas Zn, Pb, Cd, and Cr were related to domestic and agricultural wastewater” rely solely on correlations without source apportionment modelling. Similarly, the claim that heavy metals significantly affected the spatial differentiation of bacterioplankton requires multivariate analysis to separate effects from confounding variables (depth, season). The conclusion that runoff input played a significant role is not supported by quantitative runoff data. The manuscript should avoid over‑interpretation and provide evidence for causal claims.
Response 13: Thank you for your comments. We have rewritten the research conclusions in this section to avoid overinterpretation (Lines 350-353).
Comments 14: Contradiction arises between the assertion that “elemental Cd does not exhibit a clear pattern with depth” and the earlier claim that Cd is associated with agricultural wastewater. These conflicting interpretations should be reconciled with a more rigorous analysis.
Response 14: Thank you for your comments. The original statement was: Cd concentrations, on the other hand, are simultaneously controlled by multiple processes, such as the adsorption and desorption of organic matter and biological particles in water, as well as sulfide formation, and therefore do not exhibit a clear pattern of variation with depth. It mainly refers to the influence of various factors on the concentration changes of Cd in reservoir profiles, while the aforementioned correlation between Cd concentration and agricultural pollution refers to the non-point source input of surface water. The emphasis of the two sentences is different, and there is no absolute conclusion, so they are not contradictory.
Comments 15: The conclusion restates the results but does not synthesize the findings. It reiterates sources of metals and microbial dominance without linking back to hypotheses or discussing management implications. The conclusion should succinctly summarize major findings, acknowledge limitations, and suggest future research directions.
Response 15: Thank you for your comments. We have rewritten and organized the conclusion section to make it more concise and to the point (Lines 359-365).
Comments 16: References: Some cited literature appears tangential to the study. For example, the reference on “Accurate prediction of soil heavy metal pollution using an improved machine learning method” pertains to machine‑learning prediction in soils and has little relevance to reservoir water studies. Similarly, several soil-oriented papers (references [6], [7], [24], [26]) deal with terrestrial systems rather than aquatic reservoirs. Removing these and replacing them with studies on heavy-metal dynamics in reservoirs or lakes would improve the literature review.
Response 16: Thank you for your comments. Although the research objects of this literature are not reservoirs, their research methods and theories still have reference significance. The research focus of some literature is on microorganisms in water bodies or sediments, rather than nutrients and heavy metals, such as literature 26. Besides, we have deleted some of the literature and replaced it with studies on heavy metals and microorganisms in lakes and reservoirs.
Reviewer 4 Report
Comments and Suggestions for Authors
Studies on the coupling mechanism between heavy metals and nutrient salts, as well as the underlying microbial mechanisms, are relatively scarce. Which metals are related to anthropogenic activities are identified.
The methodology that was carried out is adequate, each part of the studies carried out is not described in detail, such as the type of controls of each analytical technique. In the results, a section is followed for the discussion of physicochemical parameters; before the heavy metals section. The alternatives to reduce anthropogenic discharges in the study area are not indicated.
Derived from the results and the relationship with the bacteria, which are in the study area an artificial reservoir with characteristics corresponding to the interannual regulation; where seasonal thermal stratification and seasonal hypoxia are shown. It is possible to discuss how to reduce these effects, and discuss alternatives for anthropogenic management as an option and regulation of public policies that, in other countries, have strengthened the recovery of bodies of water.
Conclusions can be improved, while at the same time better discussing suggested points to broaden discussions.
Author Response
Comments 1: The methodology that was carried out is adequate, each part of the studies carried out is not described in detail, such as the type of controls of each analytical technique. In the results, a section is followed for the discussion of physicochemical parameters; before the heavy metals section. The alternatives to reduce anthropogenic discharges in the study area are not indicated.
Response 1: Thank you for your comments. We have provided detailed supplements to the research methodology section in the revised manuscript, including specific sampling procedures, measurement programs, errors, instrument models, blank sample settings, etc (Lines 113-114, Lines 125-136, Lines 150-152). At the same time, we have improved the results and discussion sections and provided suggestions for reducing nutrient and heavy metal emissions (Lines 247-251, Lines 316-319, Lines 357-372).
Comments 2: Derived from the results and the relationship with the bacteria, which are in the study area an artificial reservoir with characteristics corresponding to the interannual regulation; where seasonal thermal stratification and seasonal hypoxia are shown. It is possible to discuss how to reduce these effects, and discuss alternatives for anthropogenic management as an option and regulation of public policies that, in other countries, have strengthened the recovery of bodies of water.
Response 2: Thank you for your comments. We have added how to improve governance measures in the revised draft, including utilizing the characteristics of water oxidation and seasonal stratification to effectively mitigate the impact of surrounding non-point source inputs on heavy metals and eutrophication in the reservoir (Lines 349-355).
Comments 3: Conclusions can be improved, while at the same time better discussing suggested points to broaden discussions.
Response 1: Thank you for your comments. We have reorganized the conclusion and discussion sections in the revised manuscript (lines 315-319, lines 349-355, lines 357-372).
Round 2
Reviewer 1 Report
Comments and Suggestions for Authors
Thanks to the authors for the corrections made.
I have no comments or questions about the content of the corrected article. I suggest it be published in this form.
The results obtained provide new information about the mechanisms and microorganisms involved in the transformation of biogenic elements and heavy metals in stratified reservoirs.
Author Response
Comments 1: Thanks to the authors for the corrections made.
I have no comments or questions about the content of the corrected article. I suggest it be published in this form.
The results obtained provide new information about the mechanisms and microorganisms involved in the transformation of biogenic elements and heavy metals in stratified reservoirs.
Response 1: Thank you for your efforts and recognition. We appreciate the valuable and scientifically significant suggestions you provided in the first round of revisions. We will continue to work hard to ensure that the article can be published as soon as possible.
Reviewer 2 Report
Comments and Suggestions for Authors
1; The revisions made are commendable. However, i have a major concern over the experimental results and responses with regard to measurements as noted her below.
(2) Although considerable efforts have been made by authors in responding to reviewer concerns, there are still major flows in the way the measurements were done.
In particular, (i) it is important to note that acidification of the water samples with nitric acid will not allow for accurate measurements of nitrate (NO3-) and and nitrites (NO2-), and in some cases in the NH4+-N. This is flawed treatment of acidified samples. (ii) in the revised MS authors in lines 129 - 132 state that USEPA has accepted recovery values of 8 -120 % without any cited reference. In any case, this is an erroneous statement. Literature does not support such an assertion. Neither has USEPA given such agreements.
(3)Other minor concerns. (i) lines 151. please remove the word "weakly' . A pH of 9.4 is not weakly alkaline.
For these two reasons, i recommend a reconsideration of the ms after major revision of the manuscript. Addressing these two concerns will make the measurements accurate in reporting, and may impact the correlations.
Comments on the Quality of English LanguageThe English is improved.
Author Response
Comments 1: The revisions made are commendable. However, i have a major concern over the experimental results and responses with regard to measurements as noted her below. Although considerable efforts have been made by authors in responding to reviewer concerns, there are still major flows in the way the measurements were done. In particular, (i) it is important to note that acidification of the water samples with nitric acid will not allow for accurate measurements of nitrate (NO3-) and and nitrites (NO2-), and in some cases in the NH4+-N. This is flawed treatment of acidified samples.
Response 1: Thank you for your comments.We fully agree with the reviewer's comments. However, the samples acidified with nitric acid were only tested for heavy metal concentration, and the original manuscript may not have expressed this clearly enough. We emphasized this point in the research methods section (Lines 117-118). In addition, we have supplemented and improved the testing methods and references for nutrients in the revised manuscript (Lines 119-125).
Comments 2: (ii) in the revised MS authors in lines 129 - 132 state that USEPA has accepted recovery values of 8 -120 % without any cited reference. In any case, this is an erroneous statement. Literature does not support such an assertion. Neither has USEPA given such agreements.
Response 2: Thank you for your comments. We fully agree with the reviewer's comments. As the previous description of the experimental method was already very detailed and specific, we have directly deleted this inappropriate expression in the revised manuscript (Lines 133-137).
Comments 3: Other minor concerns. (i) lines 151. please remove the word "weakly' . A pH of 9.4 is not weakly alkaline.
Response 3: Thank you for your comments. We have removed the word 'weakly' from the revised manuscript (Line 156).
Reviewer 3 Report
Comments and Suggestions for Authors
The manuscript has been revised and may now be considered for publication.
Reviewer 4 Report
Comments and Suggestions for Authors
The authors made suggested proposals.
The responses also improved the discussions and conclusions.
Round 3
Reviewer 2 Report
Comments and Suggestions for Authors
Authors have addressed the reviewer concerns.
The current ms is acceptable in its current form.
Comments on the Quality of English LanguageThe English is improved.
Authors have addressed the reviewer concerns.
The current ms is acceptable in its current form.